# Non-Invasive Biomarkers for Diagnosis, Risk Prediction, and Therapy Guidance of Glomerular Kidney Diseases: A Comprehensive Review

**DOI:** 10.3390/ijms25063519

**Published:** 2024-03-20

**Authors:** Lorenzo Catanese, Harald Rupprecht, Tobias B. Huber, Maja T. Lindenmeyer, Felicitas E. Hengel, Kerstin Amann, Ralph Wendt, Justyna Siwy, Harald Mischak, Joachim Beige

**Affiliations:** 1Department of Nephrology, Angiology and Rheumatology, Klinikum Bayreuth GmbH, 95445 Bayreuth, Germany; lorenzoriccardo.catanese@gmail.com (L.C.); harald.rupprecht@klinikum-bayreuth.de (H.R.); 2Kuratorium for Dialysis and Transplantation (KfH) Bayreuth, 95445 Bayreuth, Germany; 3Department of Nephrology, Medizincampus Oberfranken, Friedrich-Alexander-University Erlangen-Nürnberg, 91054 Erlangen, Germany; 4III Department of Medicine, University Medical Center Hamburg-Eppendorf, 20251 Hamburg, Germany; t.huber@uke.de (T.B.H.); m.lindenmeyer@uke.de (M.T.L.); f.hengel@uke.de (F.E.H.); 5Department of Nephropathology, Institute of Pathology, University Hospital Erlangen, Friedrich-Alexander University Erlangen-Nürnberg, 91054 Erlangen, Germany; kerstin.amann@uk-erlangen.de; 6Division of Nephrology, St. Georg Hospital, 04129 Leipzig, Germany; ralph.wendt@sanktgeorg.de; 7Mosaiques Diagnostics GmbH, 30659 Hannover, Germany; siwy@mosaiques-diagnostics.com (J.S.); mischak@mosaiques.de (H.M.); 8Department of Internal Medicine II, Martin-Luther-University Halle-Wittenberg, 06108 Halle, Germany; 9Kuratorium for Dialysis and Transplantation (KfH) Leipzig, 04129 Leipzig, Germany

**Keywords:** biomarker, CKD, kidney disease, glomerular diseases, diagnosis, risk prediction, prognosis

## Abstract

Effective management of glomerular kidney disease, one of the main categories of chronic kidney disease (CKD), requires accurate diagnosis, prognosis of progression, assessment of therapeutic efficacy, and, ideally, prediction of drug response. Multiple biomarkers and algorithms for the assessment of specific aspects of glomerular diseases have been reported in the literature. Though, the vast majority of these have not been implemented in clinical practice or are not available on a global scale due to limited access, missing medical infrastructure, or economical as well as political reasons. The aim of this review is to compile all currently available information on the diagnostic, prognostic, and predictive biomarkers currently available for the management of glomerular diseases, and provide guidance on the application of these biomarkers. As a result of the compiled evidence for the different biomarkers available, we present a decision tree for a non-invasive, biomarker-guided diagnostic path. The data currently available demonstrate that for the large majority of patients with glomerular diseases, valid biomarkers are available. However, despite the obvious disadvantages of kidney biopsy, being invasive and not applicable for monitoring, especially in the context of rare CKD etiologies, kidney biopsy still cannot be replaced by non-invasive strategies.

## 1. Introduction

Since conventional biomarkers, like estimated glomerular filtration rate (eGFR) and urinary albumin excretion rate (UAER) have their disadvantages especially in the context of early detection of chronic kidney disease (CKD), an increasing interest for new biomarkers from biofluids like blood and urine has been expressed in recent years. The ultimate goal is to identify a class of sensitive and specific biomarkers that allow for effective management of CKD, as also indicated in Figure 1. In addition to the general challenges associated with the management of CKD that are reviewed in the accompanying manuscript, effective therapy determination of the underlying etiology of CKD is of the utmost importance. Kidney biopsy in most cases still is the gold standard to determine the cause of renal insufficiency, but is associated with a significant risk of bleeding or formation of an arterio-venous fistula and is contraindicated in some patients. To establish a reliable “liquid biopsy” is therefore an important goal. A further unmet need is the detailed identification of pathophysiological pathways that lead to progressive renal function loss. Ideally, unbiased identification of biomarkers leads to the discovery of new pathways involved in the disease process and facilitates research on novel therapeutic interventions.

Here we review current data on non-invasive biomarkers to diagnose specific glomerular diseases and predict progression. We discuss data on non-invasive biomarkers in the context of specific glomerular diseases, focusing on urine proteomic and peptidomic biomarkers as this approach offers the opportunity to develop a non-invasive and unbiased diagnostic tool without a priori assumptions as to the pathogenesis of a disease. We aim to give a complete collection of blood and urine biomarkers of pathophysiological aspects of CKD and nine different categories of glomerular diseases. We refer to the most widely accepted, validated, and available markers in this review. A comprehensive overview of these markers is given in Appendix A.

## 2. Biomarkers of Glomerular Diseases

An overview of biomarkers for glomerular diseases is given in Table 1. Furthermore, information on sample medium, application, and routine clinical use is mentioned. This table is retrieved form the Appendix A which served as a basis for this review. Further information on cohorts or statistical information can be seen in Appendix A.

### 2.1. IgA-Nephropathy

IgA-nephropathy (IgAN) is the most common primary glomerulonephritis. Abnormal O-glycosylation of IgA1 represents one of the key pathogenic events in IgAN inducing a four-hit cascade, ultimately resulting in immune complex deposition in the glomerular mesangium [1].

#### 2.1.1. Routine Clinical Markers

Clinical risk factors predicting poor prognosis in IgAN consistently include time-averaged proteinuria, decreased eGFR, as well as histological lesions characterized by the Oxford classification (MEST-C score) [2,3,4]. More recently, an international risk-prediction tool based on blood pressure, proteinuria, age, race, specific pharmacotherapy, and MEST-score was made available as an online calculator [5].

#### 2.1.2. Blood Biomarkers

Galactose-deficient IgA1 (Gd-IgA1), the main autoantigen recognized by IgG or IgA autoantibodies, plays a key role in the pathogenesis of IgAN [6], leading to the formation of pathogenic immune complexes. Serum Gd-IgA1 levels, possibly in combination with Gd-IgA1-specific autoantibodies, appear to have diagnostic as well as prognostic utility representing the most promising candidate blood biomarkers for IgAN. Serum levels of Gd-IgA1 are elevated in most patients with IgAN [7] and are significantly increased compared to disease controls and healthy individuals [8], indicating the diagnostic value of the biomarker.

The prognostic value of Gd IgA1 was evaluated by Zhao et al., showing independent association with a greater risk of deterioration in renal function [9]. Elevated levels of antiglycan IgGs in the sera of patients with IgAN correlate with proteinuria [10] and may predict a faster progression to end stage kidney disease (ESKD) [11].

Yasutake et al. [12] established a novel lectin-independent method using the monoclonal antibody KM55 for measuring serum levels of Gd-IgA1. Since IgAN is very common, Suzuki et al. suggest that measurement of Gd-IgA1 and its related immune complexes may be applicable for secondary screening of examinees with hematuria in general checkups [13].

#### 2.1.3. Urine Biomarkers

The association of Dickkopf-3 (DKK3) with loss of eGFR has been studied specifically in patients with IgAN in the STOP IgAN trial [14]. DKK3 was significantly associated with eGFR loss, and DKK3 levels above the median of 779 pg/mg creatinine were associated with a mean eGFR decline of 19.1% during the run-in phase of the study [15]. Rising DKK3 during the treatment phase of STOP IgAN was associated with a significant eGFR decline, whereas stable or decreasing urinary DDK3-to-creatinine levels indicated a more favorable course of kidney function [15].

In recent years, urine proteomics has been applied to serve as a diagnostic tool in patients with IgAN. In a study by Haubitz et al. [16], total peptide pattern in human urine was investigated. Polypeptide patterns in patients with IgAN were compared with patterns established in patients with several other kidney diseases, as well as in healthy volunteers. A total of 22 peptides with the highest discriminatory values between IgAN and healthy individuals were identified, and support vector machines were employed to enable the classification of patients based on all 22 polypeptides simultaneously. Correct classification of patients versus healthy controls was achieved with a sensitivity of 100% and a specificity of 90% after cross-validation. Similar sets of peptides were generated to discriminate between IgAN and other forms of glomerular diseases. The same method was later applied to the prediction of disease progression in 209 patients with IgAN [17]. Progression was defined by eGFR slope, and tertile comparison was performed. In total, 237 peptides showed significant differences in abundance, and these included apolipoprotein C-III, alpha-1 antitrypsin, different collagens, fibrinogen, titin, hemoglobin subunits, uromodulin, and polymeric Ig receptor. An algorithm based on these protein fragments (IgAN237) and clinical data showed significant added value for the prediction of IgAN progression, as compared with clinical parameters alone (area under the curve (AUC) 0.89 versus 0.72). IgAN237 was recently validated in an independent set of prospective collected urine samples of 103 patients with biopsy-proven IgAN [18]. Patients were grouped into non-progressors and progressors according to their urinary peptidomic profile. In multiple regression analysis, a baseline label of progressor or non-progressor according to IgAN237 was an independent predictor of eGFR decline.

### 2.2. Minimal Change Glomerulonephritis

Minimal change glomerulonephritis (MCGN) and focal segmental glomerulosclerosis (FSGS) are seen as a continuum of the same pathogenic process by some in the field, whereas others consider these conditions as two completely separate disease entities. As long as this dispute is not finally resolved, we will report on MCGN and FSGS as two different entities and try to elaborate differences between them with regard to biomarkers.

#### 2.2.1. Routine Clinical Markers

MCGN usually presents with nephrotic syndrome of sudden onset, and especially in childhood is characterized by a very good response to steroid treatment. Steroid dependence or steroid resistance are markers of a worse kidney prognosis.

#### 2.2.2. Blood Biomarkers

A very important step in understanding the pathogenesis of MCGN was made by Watts et al. [19]. With knowledge from rodent models, the authors hypothesized that nephrin autoantibodies may be present in patients with MCGN. It could be shown that 18 of 62 adults and children (29%) with MCGN in the NEPTUNE study cohort had detectable nephrin autoantibodies by ELISA testing, while 98% of nephrotic control patients with anti-hPLA2R antibodies were negative for anti-nephrin antibodies. Data indicated this as having a role as a biomarker for active disease and shorter relapse-free survival. Further studies need to corroborate these findings, but anti-nephrin autoantibodies might turn out to be an excellent biomarker for the diagnosis of MCGN in at least a subset of these patients.

In an effort to predict steroid resistance in childhood nephrotic syndrome, Agrawal et al. performed proteomic analysis on paired plasma samples obtained at disease presentation before glucocorticoid initiation and after 7 weeks of therapy. They identified a biomarker panel consisting of vitamin D binding protein, adiponectin, and matrix metalloproteinase 2 with a significant ability to predict steroid resistant nephrotic syndrome at disease presentation (AUC = 0.78) [20].

#### 2.2.3. Urine Biomarkers

There is some evidence that urinary cluster of differentiation 80 (CD80) may be useful in discriminating between MCGN and other glomerular diseases including FSGS [21]. In a study with 411 patients from the Mayo Clinic and the NEPTUNE network, CD80 excretion was highest in patients with MCGN [22]. In a small study in biopsy-proven patients with MCGN (*n* = 17) or FSGS (*n* = 22), a significant increase in urinary CD80 excretion was found in patients with MCGN in relapse compared to those in remission or to those with FSGS [23]. In line with this, significantly increased urinary CD80 levels were reported in MCGN patients in relapse, when compared to patients in remission, FSGS patients, or control subjects [24].

### 2.3. Focal Segmental Glomerulosclerosis

FSGS is a descriptive renal histologic lesion with diverse causes and pathogenicities that are linked by podocyte injury and depletion and lead to significant proteinuria. The lesion of FSGS can be subdivided into genetic, primary (idiopathic), and secondary forms. The therapeutic approaches to the various forms of FSGS vary considerably. Therefore, it is crucial to establish diagnostic algorithms that can reliably distinguish between the different forms, especially between primary and secondary FSGS, to avoid unnecessary and not to withhold necessary immunosuppressive-based therapies.

#### 2.3.1. Routine Clinical Markers

Primary FSGS usually presents with nephrotic-range proteinuria (>3.5 g/day) with complete nephrotic syndrome, hypertension, and microhematuria [25], and a rapid onset of disease. Effacement of the epithelial foot processes of glomerular podocytes is thought to be diffuse in primary FSGS [26,27,28]. Secondary FSGS patients can present with a broad range of proteinuria (including nephrotic range), but in general do not develop complete nephrotic syndrome. Proteinuria frequently shows a slow increase over time [29]. Risk factors like obesity, vesicoureteral reflux, renal agenesis, reduced nephron mass, or infection may be present.

While the degree of baseline proteinuria, remission of proteinuria, baseline eGFR, and histopathological findings can help estimate prognosis, disease-specific routine markers are missing [30,31,32].

#### 2.3.2. Blood Biomarkers

The search for plasma biomarkers has focused on the putative permeabilizing factor. With several other proposed markers, cardiotrophin-like cytokine factor 1 (CLCF1) and soluble urokinase plasminogen activator-receptor (suPAR) [33] appeared to be the most promising ones.

CLCF1 belongs to the interleukin-6 family. High levels were reported in the active plasma fraction of patients with recurrent FSGS compared to controls [33], and treatment of mice with recurrent doses of CLCF1-induced FSGS-like lesions [34].

SuPAR was found to be increased in the serum of primary FSGS patients [35]. Injection of recombinant uPAR into a knockout mouse model induced proteinuria [35]. Unfortunately, later studies were unable to reproduce these results in independent cohorts [36,37,38]. It was also shown that suPAR levels increased with decreasing eGFR. In fact, suPAR levels above the suggested cutoff for FSGS were found in 88% and 95% of non-FSGS patients with an eGFR between 30–45 mL/min or <30 mL/min, respectively [39]. Due to these conflicting results, it seems unlikely that suPAR represents the main permeabilizing factor in FSGS.

#### 2.3.3. Urine Biomarkers

One biomarker that has been specifically associated with FSGS, in particular with post-transplantation recurrent FSGS, is apolipoprotein A-Ib (apoA-Ib), a modified form of Apo-I. In a total of 119 patients studied, the Apo-Ib form was detected in the majority of relapse patients, but not in relapse-free patients of the control group [40]. Kalantari tried to establish a urinary biomarker for steroid-resistant and steroid-sensitive FSGS using high resolution mass spectrometry and identified 21 proteins as discriminating species. The most drastic fold changes were observed for ApoA-I and matrix-remodeling protein 8 [41].

Several miRNAs are upregulated in glomeruli of primary FSGS patients [42], and one of them, miRNA-193a, might be relevant for development of the disease, as it decreases the expression of WT-1, which compromises podocyte function [43]. Interestingly, miRNA-193a is also increased in urinary exosomes of children with primary FSGS, compared to MCGN patients [44], which might make this marker amenable for urinary diagnostics. The validity of this marker to differentiate between primary and secondary FSGS was, however, not investigated.

We recently developed a urine peptide-based classifier of 19 pFSGS and 44 sFSGS patients. The classifier consists of 93 peptides, discriminates pFSGS from healthy controls and other CKD etiologies with a specificity of 99.1% and 94.7%, respectively, and is able to discriminate pFSGS from forms of sFSGS with a specificity of 100% and a sensitivity of 84.2%. The classifier resulted in an area under the receiver operating curve of 0.95 [45]. Many of the peptides in this classifier were fragments of different collagens, and most of them were reduced in pFSGS. Upregulation was observed, among others, in apolipoprotein A-I (see above) and complement C3 peptides.

### 2.4. Membranous Nephropathy

Among several autoantigens associated with membranous nephropathy (MN), phospholipase-A2-receptor-1 (PLA2R1) and thrombospondin-type1-domain-containing-protein-7A (THSD7A) are the most frequent and pathogenicity-proven antigens in primary MN [46,47]. Such autoantibodies can serve as sensitive and specific biomarkers and offer the unique opportunity of diagnosing MN on a molecular level, non-invasive disease monitoring, and even biopsy-independent diagnosis in patients with contraindications to kidney biopsy.

#### 2.4.1. Routine Clinical Markers

The clinical course of MN exhibits large variability, with approximately 1/3 of patients experiencing spontaneous remission, while 1/3 of patients reach ESKD within ten years [48,49]. Clinical features such as age at disease onset >60 years, male sex, persistent proteinuria > 8 g/day, and impaired eGFR serve as risk stratification of severe disease development. Spontaneous remission is predicted by a proteinuria decrease > 50% compared to baseline during the first year [49].

#### 2.4.2. Blood Biomarkers

PLA2R1 autoantibodies play an outstanding role in diagnosis and disease monitoring in PLA2R-positive MN. Their levels are associated with spontaneous remission, therapy response, and progressive loss of kidney function [50,51,52]. Decreasing PLA2R1 antibodies are followed by a decrease in proteinuria and reemerging or increasing PLA2R1 antibodies precede relapse and relapse after kidney transplantation [52]. The diagnosis of MN based on serum PLA2R1 antibodies alone is discussed controversially. A meta-analysis concluded that PLA2R1 antibodies have high diagnostic value, but study heterogeneity is due to several factors, especially test intervals [53]. Specificity of PLA2R1 antibody measurement is comparable to the diagnostic outcome of a kidney biopsy in patients with nephrotic syndrome and might ultimately render kidney biopsy dispensable in a subgroup of patients [54]. For therapy guidance, KDIGO guidelines recommend a cut-off of anti-PLA2R1 antibodies >50 RU/mL as a high-risk factor for progressive loss of kidney function in patients with impaired eGFR, high proteinuria, or incomplete response to conservative antiproteinuric therapy. Immunosuppressive treatment is advised in such cases [54]. Depending on PLA2R1 titers, antibody measurement is recommended every 3–6 months [54].

Due to its lower frequency, the diagnostic and prognostic relevance of anti-THSD7a titers is less clear. However, in light of the experimentally proven pathogenicity, similarities in pathomechanisms, and (sparse) clinical data, chances are high that anti-THSD7a titers perform on a comparable level to PLA2R1 [46,55]. In 49 patients with THSD7a-positive MN anti-THSD7a titers correlated with disease activity, therapy response, and remission, with high baseline titers predicting poor clinical outcome [55].

Antibodies against semaphorin-3B (Sema3B) are seldom, but predominantly found in young children, offering the potential for a non-invasive diagnosis of MN in pediatric patients to avoid kidney biopsy [56]. Patients with antibodies against neuronal-antigen-neural-epidermal-growth-factor-like-1-protein (NELL-1) are prone to have a concurrent malignancy, implicating the need for cancer screening [57,58].

The autoantigen exostosin-1/2 (EXT1/2) is one of a few associated with membranous lupus nephritis (LN). In a retrospective study, about 32% of patients were histologically positive for EXT1/2 [59], which was associated with fewer histological chronic changes and a lower rate of progression to ESKD.

Numerous other autoantibodies associated with MN were recently described and hold the promise to serve as non-invasive and/or histological biomarkers (see SI). However, they require further validation and are not yet available for routine clinical testing.

Cremoni et al. studied cytokine profiles and found an association between high IL17A levels in MN patients and increased thromboembolic events and relapse [60]. Cantarelli et al. found increased TNF-alpha levels and autoreactive circulating plasma cells correlate with anti-PLA2R1 levels and disease activity, which was investigated further in a phase 2 clinical trial on the effect of anti-CD38 (Felzartamab) targeting plasmablasts and plasma cells in PLA2R1-positive MN [61]. Increased TNF-alpha levels decreased upon anti-CD20 therapy in a study by Rosenzwajg et al. [62]. Additionally, they found lower Tregs at baseline and a significant increase upon anti-CD20 therapy, suggesting Treg population dynamics as an early predictor of response to Rituximab [62].

#### 2.4.3. Urine Biomarkers

Urinary beta-2-microglobulin (B2MG) was evaluated to stratify patients for immunosuppressive treatment [63]. B2MG predicted disease progression in primary MN with 88% sensitivity and 91% specificity [63]. No prediction of response to immunosuppression by those urinary protein levels alone was shown, but including urinary B2MG, alpha-1-microglobulin, and IgG for risk assessment of progressive loss of kidney function in MN patients is recommended by KDIGO guidelines [54]. However, elevated urinary B2MG can have many causes, and does not serve as a specific marker. More recent discoveries of potential biomarkers result from differential urinary proteomics and metabolomics but lack widespread validation so far.

### 2.5. Membranoproliferative Glomerulonephritis/C3-Glomerulopathy

Membranoproliferative glomerulonephritis (MPGN) describes a pattern of glomerular injury characterized by an increase in mesangial cellularity and matrix with thickening of the glomerular capillary walls secondary to subendothelial and in part subepithelial deposition of immune complexes and/or complement factors, and new basement membrane formation. MPGN has been reclassified using the immunohistological/immunofluorescence findings into immunoglobulin-mediated disease (IC-MPGN; driven primarily by the classical complement pathway), versus non-immunoglobulin-mediated disease (C3-glomerulopathy (C3GP); driven by an overactivity of the alternative pathway) [64,65]. C3GP is further differentiated into dense deposit disease (DDD) and C3-glomerulonephritis (C3GN). There is, however, major overlap between these entities both in terms of clinical as well as laboratory and morphological characteristics.

#### 2.5.1. Routine Clinical Markers

Reduction in the serum concentration of complement components C3 or C4 is frequently observed. While in immune-complex-mediated forms of MPGN variable reduction in both C3 and C4 is observed (activation of the classical pathway), C3GN is often characterized by low C3 but relatively normal C4 levels (predominant activation of the alternative pathway; low C3 in 40–60%, low C4 in 0–4.5%) [66]. Since there is a paucity of larger cohorts of MPGN patients that are already reclassified according to the new criteria, no applicable routine clinical prognosticators have been identified in longitudinal studies.

#### 2.5.2. Blood Biomarkers

Changes in specific factors of the complement pathway that may lead to overactivity of the alternative pathway were detected in C3GN. These factors include acquired autoantibodies that stabilize the C3 convertase, for example, C3Nef, C4Nef, or anti-CFB antibodies, or that block the action of pathway inhibitors like anti-FH and anti-FI antibodies [64,67,68,69].

In a multivariate analysis of a French nationwide cohort of 165 patients, age, eGFR < 60 mL/min, and the presence of rare disease-predicting variants in complement genes predicted risk of progression to kidney failure. In addition, the complement biomarker profile of normal C3/high sC5b-9 or low C3/normal sC5b-9 was independently associated with worse kidney prognosis [70].

#### 2.5.3. Urine Biomarkers

Studies that were specifically aimed at the identification of urinary biomarkers in MPGN or C3GP are still lacking. Urinary sC5b-9 correlated with increased plasma levels and proteinuria in the study by Podos et al. [71]. In another approach, mass spectrometry based peptidomics data from the Human Urinary Proteome/Peptidome Database were extracted. In total, 23 peptides could be identified, originating from the complement proteins C3, C4, and CFB. Significantly increased excretion of C3-derived peptides was observed in MCGN, FSGS, MN, LN, IgAN, MPGN, and C3G, with the highest excretion found in MCGN [72]. Increased excretion of C3 peptides is therefore of no use to diagnose MPGN or C3GP. On the other hand, excretion of FB peptides was highest in MPGN but was not elevated in other glomerular diseases including C3GP.

### 2.6. Fibrillary Glomerulonephritis

Fibrillary glomerulonephritis (FGN) is a rare glomerular disease defined by glomerular deposition of Congo red-negative, randomly oriented straight fibrils [73]. It should not be confused with the even rarer immunotactoid glomerulopathy, which is characterized by the deposition of microtubules composed of mostly monoclonal proteins [74].

Recently, a novel tissue biomarker of FGN, DNAJ homolog subfamily B member 9 (DNAJB9), has been identified [75,76,77]. DNAJB9 belongs to a family of proteins that function as “co-chaperones” to heat-shock protein 70 (hsp-70). It is expressed in all healthy tissues and is localized to endoplasmic reticulum (ER), and is upregulated by inflammatory mediators. DNAJB9 immunohistochemistry has a 98% sensitivity and >99% specificity for FGN and has become the gold standard in the diagnosis of FGN [75].

#### 2.6.1. Blood Biomarkers

The group of researchers that discovered the role of DNAJB9 immunohistochemistry in the diagnosis of FGN also detected a 4-fold higher abundance of serum DNAJB9 in FGN patients when compared to controls [78]. Serum DNAJB9 levels accurately predicted FGN with moderate sensitivity (67%), high specificity (98%), and a positive and negative predictive value of 89% and 95%, respectively [78].

#### 2.6.2. Urine Biomarkers

Urine DNAJB9 has not been investigated so far and its role as a potentially useful non-invasive biomarker in the future remains unclear.

### 2.7. Antineutrophil Cytoplasmic Autoantibody-Associated Vasculitis

#### 2.7.1. Routine Clinical Markers

Approximately 90% of antineutrophil cytoplasmic antibody (ANCA)-associated vasculitis (AAV) are ANCA-positive [79], but up to 10% of patients can be ANCA-negative [80]. The predictive value of positive ANCA for diagnosis of AAV depends primarily upon the clinical presentation of the patient. In patients with rapidly progressive glomerulonephritis (RPGN) at presentation, the presence of ANCAs will predict AAV with very high accuracy, while their accuracy is significantly lower in patients with less compelling symptoms. Furthermore, specificity for AAV is limited as ANCA detection has also been reported in a variety of other diseases, and false positive results in immunofluorescence can be caused by several other antigens [81,82].

The role of ANCAs as a disease activity marker has been the subject of debates, and study outcomes are not conclusive in this regard [83]. A rise in ANCA levels has been associated with increased risk of relapse [84], however, a rise in ANCA titers is not always followed by a relapse [85], and an ANCA rise cannot automatically be used to initiate preemptive treatment since less than half of the patients with an ANCA rise will experience a relapse within a year [86].

#### 2.7.2. Blood Biomarkers

Neutrophil extracellular traps (NETs), composed of DNA, histones, and neutrophil proteins, are released by ANCA-stimulated neutrophils and contain the targeted autoantigens PR3 and MPO. Deposition of NETs in inflamed kidneys and circulating MPO–DNA complexes suggest that NET formation triggers vasculitis and promotes the autoimmune response against neutrophil components in AAV [87]. Abnormal formation and/or insufficient clearance of NETs may contribute to increasing levels of cell-free DNA (cfDNA) in granulomatosis with polyangiitis. cfDNA levels or NETs may serve as a marker of disease activity in AAV [88,89].

Calprotectin is expressed in neutrophils, monocytes, and early differentiated macrophages, and is secreted locally at the site of inflammation by phagocytes. Serum calprotectin levels were increased after discontinuation of AAV treatment and significantly elevated in relapse [90]. In the RAVE population, rising calprotectin levels during treatment in a subgroup of PR3-ANCA patients were indicative of a higher risk of relapse by 18 months [91]. Serum calprotectin may assist in identifying those patients requiring more intensive or prolonged treatment.

Serum levels of complement C5a and levels of fragment Bb were higher in patients with active AAV compared with AAV in remission or healthy controls [92,93]. The important role of C5a and the neutrophil C5a-receptor (C5aR) as an amplification loop for ANCA-mediated neutrophil activation has been described [94], and has recently led to the development and approval of a new therapeutic intervention of C5aR inhibition with avacopan [95].

In a prospective evaluation from patients enrolled in the RAVE trial, the three chemokines matrix metalloproteinase 3 (MMP-3), tissue inhibitor of metallopeptidase inhibitor 1 (TIMP-1), and B-lymphocyte chemoattractant chemokine (C-X-C motif) ligand 13 (CXCL13) were able to distinguish active disease from remission with an AUC of 0.8 and a likelihood ratio of 4.3–4.6, which makes them promising candidates for further evaluation [96].

#### 2.7.3. Urine Biomarkers

Urinary monocyte chemotactic protein 1 (uMCP-1) is a very promising biomarker to discriminate between active kidney disease and remission, and to monitor treatment [97,98].

Another promising biomarker is urinary soluble CD163 (usCD163), a soluble scavenger receptor for the hemoglobin–haptoglobin receptor complex. Patients with active vasculitis had markedly higher usCD163 levels than patients in remission, disease controls, or healthy controls [99,100,101]. In a recent trial in 47 patients with ANCA-associated glomerulonephritis, usCD163 concentrations were elevated at disease onset in all patients with active renal vasculitis and undetectable among patients in remission [102]. Therefore, usCD163 seems to represent a useful biomarker for the detection of active renal vasculitis, relapse, and for monitoring treatment response.

Finally, urine proteomics with capillary electrophoresis-mass spectrometry using 18 distinct biomarkers showed a high discrimination for AAV patients from other kidney diseases, and represents a promising tool supporting the non-invasive identification of patients with active AAV [103].

### 2.8. Lupus Nephritis

LN is a common and often the earliest organ manifestation in systemic lupus erythematodes (SLE). The mortality is higher in patients with LN than in those without, being as high as 25% among those with severe proliferative forms of the disease [104]. Despite new treatments, 10 to 30% of LN patients still progress to ESKD [105,106].

#### 2.8.1. Routine Clinical Markers

According to international guidelines, anti-dsDNA antibodies, C3 and C4 complement levels, eGFR, urinalysis with urine sediment microscopy, and proteinuria can help monitor LN with limited accuracy. Because clinical and histopathological features of inflammation or remission often do not correlate [107], the need for better biomarkers is high.

#### 2.8.2. Blood Biomarkers

The production and presence of autoantibodies is an important and defining hallmark of SLE, and thus autoantibodies to double-stranded DNA (dsDNA) and C1q as well as antinuclear antibodies (ANA) and anti-nucleosome antibodies have been used as biomarkers. In a review by Yin et al., the accuracy of anti-C1q in the diagnosis of LN in patients with SLE was evaluated in the Chinese population. The pooled sensitivity was 0.58, the pooled specificity 0.75, and the SROC-AUC was 0.794 [108].

#### 2.8.3. Urine Biomarkers

Complement components can be found in the urine of LN patients and may be indicative of complement activation within the kidney and reflect active kidney disease. It was shown that urine C3d levels were superior to plasma C3, C4, C4d, C5b-9, and anti-dsDNA to differentiate acute from chronic LN [109]. Negi et al. reported elevated urine C3d levels only in patients with active LN as compared to inactive LN and non-renal SLE [110]. Urinary C3d could serve as a potential biomarker to determine LN disease activity.

In an overview of systematic reviews by Guimaraes et al., the biomarkers with the best accuracy profile were urinary tumor necrosis factor-like weak inducer of apoptosis (uTWEAK), a member of the TNF superfamily, urinary monocyte chemoattractant protein-1 (uMCP-1), and urinary neutrophil gelatinase-associated lipocalin (uNGAL), which were more sensitive than specific for most of the analyzed outcomes [111]. SLE patients with active kidney disease have significantly higher urinary TWEAK levels than SLE patients without active kidney disease [112]. One systematic review analyzed the use of uMCP-1 in detecting kidney disease activity [113]. The pooled sensitivity was 0.89, the pooled specificity 0.63, and the SROC-AUC was 0.90. uMCP1 levels are significantly higher in patients with LN compared with healthy controls [114], and urinary concentration has been related to a decrease in renal function [115]. Furthermore, uMCP1 may increase between 2 and 4 months before renal relapse and may be a valuable prognostic factor [116]. uMCP1 seems to be superior to conventional serological biomarkers, but high levels are not specific for LN as they have also been reported in diabetic kidney disease, IgAN, MN, FSGS, and polycystic kidney disease [117]. Urinary EGF correlated with histologic kidney damage in LN patients and adverse long-term kidney outcomes [118].

Soluble urinary VCAM-1 showed a strong association with the presence of LN, with clinical and histological activity indices, and with more severe renal lesions [119]. In addition to VCAM-1, the activated leucocyte cell adhesion molecule (ALCAM) was measured in a cohort of active and inactive LN patients. VCAM-1 and ALCAM were elevated in active LN as compared to inactive LN or healthy controls. ALCAM levels were higher in the proliferative classes III and IV of LN, and VCAM-1 was indicative of a long-term loss in renal function [120,121].

In another attempt to predict the activity of LN, Brunner et al. performed an assessment of 16 urinary biomarkers in pediatric SLE patients. Of these, NGAL, MCP-1, CP, adiponectin, hemopexin, and KIM-1 were combined to generate the renal activity index for lupus (RAIL). These urinary biomarkers predicted the LN activity state (NIH-AI) with over 92% accuracy. Therefore, RAIL might represent an interesting non-invasive tool to determine LN activity [122].

Finally, urinary proteomics has led to the identification of prostaglandin H2D-isomerase (PGDS), alpha-1 anti-chymotrysin (ACT), haptoglobin, and retinol binding protein (RBP) as biomarkers of active LN [123,124].

To develop prediction models for one-year treatment response, a machine learning approach combining traditional clinical data and novel urine biomarkers was successfully established [125]. For model development, 13 urinary biomarkers for LN that were ranked in the top 50% in an unbiased PubMed search were selected. The prediction of treatment response after one year was significantly improved by including urinary biomarkers in addition to clinical markers (AUC for support vector machine model improved from 0.761 to 0.841).

### 2.9. Diabetic Kidney Disease

Diabetic kidney disease (DKD), a leading cause of CKD and ESKD worldwide, affects about 30% of type 1 diabetes (TDM1) and about 40% of type 2 diabetes (TDM2) patients [126]. Determining the true prevalence of DKD is difficult as clinical diagnosis often relies on eGFR decline and/or albuminuria with pre-existing diabetes but without histological diagnosis [126,127]. Inflammation has been recognized as a major contributor to DKD during the last decade, offering the potential for establishing new biomarkers and therapies in DKD126.

#### 2.9.1. Routine Clinical Markers

Albuminuria (>30 mg/g creatinine) and an eGFR < 60 mL/min/1.73 m^2^, together with clinical features such as diabetes duration and diabetic retinopathy, are currently routinely used as clinical markers for the DKD diagnosis and deterioration of kidney function [126]. However, kidney disease, especially in type 2 diabetes mellitus (T2DM), is too complex for assessment with albuminuria and eGFR alone [128]. Albuminuria has many limitations, like high variability, low sensitivity, or specificity for accurately predicting kidney function, and can be reversible [129]. Kidney function loss can occur even in the absence of elevated albuminuria and may subsequently progress to ESKD [130].

#### 2.9.2. Blood Biomarkers

Two of the recently emerged most promising inflammation-related biomarkers displaying a significant association with disease initiation and progression are tumor necrosis factor receptor-1 and -2 (TNFR1, -2). TNF-α, a central pleiotropic proinflammatory cytokine, plays a pivotal role in inflammation and apoptosis and interacts with its two distinct cell surface receptors, TNFR1 and TNFR2, also found as soluble forms (sTNFR1, sTNFR2) in serum [131].

Several studies have demonstrated that circulating TNFR levels, especially sTNFR1, predict DKD progression and ESKD in patients with type 1 diabetes mellites (T1DM) and T2DM from diverse populations [132,133], summarized in [131]. They outperform or improve the power of albumin creatinine ratio (ACR) in predicting disease progression [134,135], and additionally show promise as potential markers for identifying patients at a high risk of developing DKD [136]. Furthermore, TNFRs could also serve as stratification markers for patients at a high risk of progression to improve enrichment in clinical trials [137]. Recently, a set of 17 circulating inflammatory proteins enriched in members of the TNF receptor superfamily, the kidney risk inflammatory signature (KRIS), was identified as having a strong association with the development of ESKD [138].

Besides TNFRs, other inflammatory biomarkers, such as the acute phase C-reactive protein (CRP) and the proinflammatory cytokine interleukin-6 (IL-6) have been linked to the development and progression of DKD. Different studies in patients with T1DM and T2DM show an association of high-sensitivity CRP (hs-CRP) levels with DKD [139,140], a positive correlation with ACR [140,141], and a predictive role of serum hs-CRP in estimating the risk of DKD in individuals with T2DM [142]. Increased serum levels of IL-6 have been found in patients with DKD compared with diabetics without kidney disease [143], which were independently associated with an increased risk for disease progression [144].

DKD is a progressive disease with structural changes and damage to the glomeruli and tubules. Several kidney injury biomarkers, such as the kidney injury marker 1 (KIM-1), have emerged as potential prognostic candidates.

In T1DM patients, elevated baseline plasma KIM-1 showed a strong association with risk of early progressive kidney function decline in patients with normal kidney function at baseline [145]. In individuals with T1DM and proteinuria with longitudinal follow-up, blood KIM-1 levels at baseline predicted eGFR loss and ESKD risk during long-term follow-up [146]. Similar associations are seen in T2DM patients from various studies [147,148]. A recent study of T2DM individuals and incident or early DKD identified the protein marker KIM-1 as the most relevant predictor of eGFR trajectory, with baseline eGFR as an important clinical covariate [149].

#### 2.9.3. Urine Biomarkers

In recent years, various potential urinary markers for DKD have been identified including NGAL, a small circulating protein belonging to the lipocalin protein family, liver-type fatty acid-binding protein (L-FABP), a small protein involved in the metabolism of long fatty-chain acids, or proteomics markers such as CKD273 [150,151].

In diabetics with microalbuminuria, a positive correlation between increased NGAL levels and ACR has been reported [151,152], as well as an association of increased urinary NGAL levels with kidney function [152,153]. Two recent meta-analyses [139,152] support the potential diagnostic value of urinary NGAL for DKD classification.

Beside injury and inflammatory markers, the urinary CKD273 classifier, a proteome-based classifier consisting of 273 peptides, has emerged as a promising candidate for diagnostic and prognostic purposes in DKD. Based on the peptides contained in this classifier, it is reasonable to assume that it mostly depicts fibrosis and inflammation, processes that are thought to represent the initial molecular changes leading to DKD. Different cross-sectional studies demonstrated a predictive value of urinary CKD273 for the development and worsening of albuminuria in patients with T2DM [154,155]. Additionally, it was shown that CKD273 might be a good predictor for eGFR decline and incidental risk for CKD (GFR < 60 mL/min/1.73 m^2^) in T1DM and T2DM [156,157].

## 3. Synopsis and Clinical Applicability

Current standards in the diagnosis and management of glomerular diseases rely heavily on invasive kidney biopsy, and KDIGO guidelines for glomerular diseases [54] revolve mainly around histological information. Regarding the current magnitude and speed of non-invasive biomarker research and considering some examples of glomerular diseases (MN, AAV) where non-invasive biomarkers have in part replaced kidney biopsy, we believe that the role of invasive diagnostic tools like kidney biopsy could become smaller due to risks and costs of the procedure, in the case where robust and reliable tools based on non-invasive diagnosis become widely available. We therefore want to provide a theoretical decision tree including current research findings for scenarios where kidney biopsy might not be possible or feasible.

Many of the non-invasive diagnostic and prognostic tools available today are ready to be used and can support diagnosis, prognosis, and selection of therapeutic strategies (Figure 2). Patients with kidney disease generally present to nephrologists with indicative clinical patterns, carrying the potential to enable definite diagnoses on the basis of these scenarios which can be augmented with non-invasive biomarker-guided diagnostic schemes.

As a first step, important diseases can be ruled out or confirmed by history, imaging tools, and the presence of diabetes. Acute kidney injury (AKI) from circulatory causes can be confidently diagnosed based on disease history. Signatory imaging features enable diagnosis in autosomal dominant polycystic kidney disease (ADPKD), and obstructive kidney disease including congenital anomalies of the kidneys and genitourinary tract (CAKUT). DKD cannot automatically be diagnosed based on the combination of diabetes and kidney disease. A non-invasive option to enhance certainty on presence or absence of DKD in diabetics is the urinary proteomic pattern indicative of DKD.

In cases of morphologically inapparent kidney disease in the absence of diabetes, the next step is to determine excretory renal function and proteinuria. In the presence of a predominantly glomerular, nephrotic, and non-selective proteinuria, MCGN, FSGS, renal amyloidosis, MN, or MPGN/C3GP should be considered. Renal amyloidosis can be excluded or confirmed by the presence of characteristic ratios of lambda and kappa light chains. MN can also be detected by characteristic autoantibodies described in detail above. If neither a diagnostic light chain profile nor antibodies relating to MN is found, MCGN or focal glomerulosclerosis is highly suspected. The role of nephrin antibodies in the diagnosis of MCGN needs to be established. Further support of the diagnosis can be obtained from the differential diagnostic urinary peptide patterns introduced by Siwy et al. [158] and further refined by Mavrogeorgis et al. [159].

Inflammatory glomerular diseases are characterized by a pathological erythrocyte passage of the glomerular basement membrane with the appearance of dysmorphic erythrocytes in the urine. Further discrimination can be obtained using highly distinctive proteomic urinary or genomic patterns. An exemplary kidney disease is Alport’s syndrome, which can be initially detected by the assessment of urine cellular swaps (dysmorphic erythrocytes) in conjunction with known mutations in collagen IV. One other condition is IgAN, the most common glomerulonephritis. Genetic diagnostics for this disease—in contrast to Alport nephropathy—is not well established, but IgAN can be detected with a high degree of certainty using a proteomic urinary pattern (IgAN 237).

A rapidly progressive loss of kidney function in conjunction with proteinuria and dysmorphic erythrocyturia suggests diseases of the entire glomerular compartment with extracapillary proliferation. Goodpasture syndrome can be diagnosed through the detection of anti-glomerular basal membrane antibodies. In autoimmune vasculitis, the detection of ANCAs is diagnostic, while LN can be diagnosed using antinuclear and dsDNA antibodies. We are now in the fortunate position of being able to assess the diagnostic reliability of biomarkers, genes, and proteomic patterns in many kidney diseases based on the long-term gold standard of histomorphological techniques, which remain mandatory for not yet well-distinguished diseases. As an example, membranoproliferative (complement-dependent) GN is not characterized via a certain biomarker pattern, and still holds difficulties with immunohistological diagnostics as well. These types of entities should be the focus of temporary diagnostic algorithm development, which nowadays must include next-level data integration by both human and artificial intelligence.

The first steps towards a non-invasive liquid kidney biopsy have been taken. The technology can be already applied today in patients that are not candidates for conventional biopsy.

## 4. Conclusions

In this review, we give an overview of biomarkers of glomerular kidney diseases which have been implemented in clinical practice or are promising candidates for future clinical implementation. We provide a clinical decision tree for non-invasive diagnosis and management of glomerular diseases in cases of kidney biopsy not being available or feasible. Most of the biomarkers are not yet incorporated in KDIGO guidelines on the management of glomerular diseases, in some cases independent validation in larger cohorts is missing. Due to the downsides of invasive kidney biopsy, the need for biomarkers in glomerular diseases is high and encouraged by KDIGO. Future biomarker research using large-scale approaches like proteomics, transcriptomics, and metabolomics seem to be most promising. Furthermore, untargeted approaches and unsupervised machine learning using AI might offer new perspectives to biomarker panels or classifications of specific glomerular diseases like MCGN or FSGS.

## Figures and Tables

**Figure 1 ijms-25-03519-f001:**
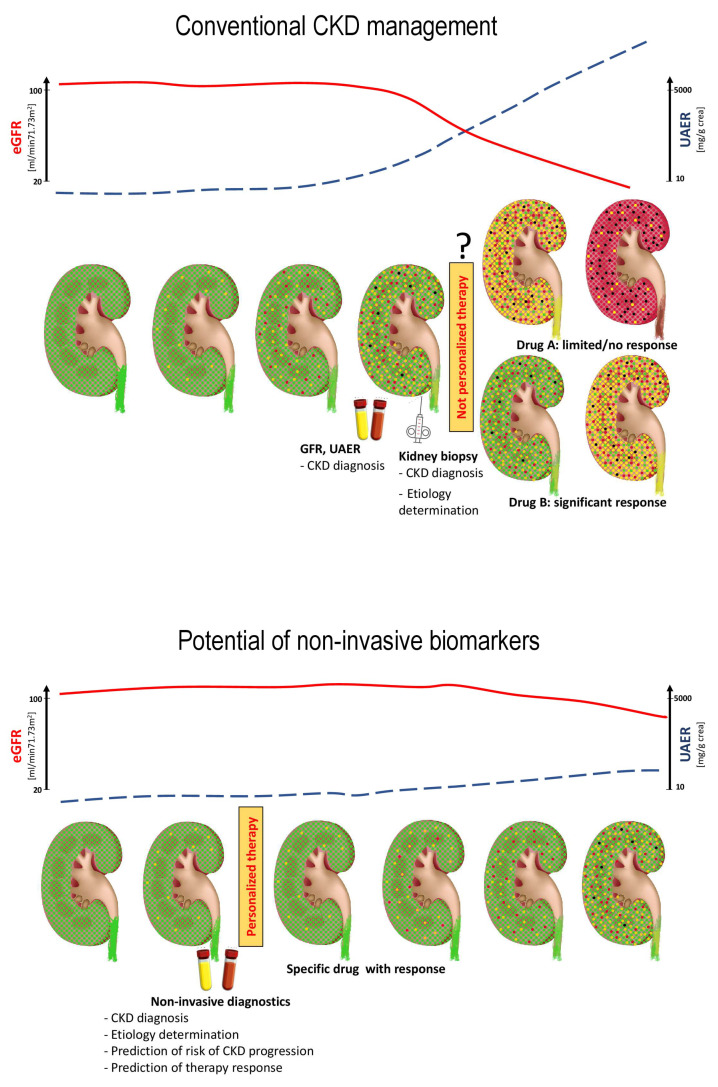
Schematic concept of chronic kidney disease (CKD) onset and progression by conventional CKD management (upper panel), and the potential of new non-invasive biomarkers (lower panel). The conventionally used biomarkers like estimated glomerular filtration rate (eGFR, red line) and urinary albumin excretion rate (UAER, blue line) allow identification of CKD in intermediate or advanced stages with nephrons harboring pathological molecular changes (indicated by yellow), already irreversibly damaged (indicated by red), or even destroyed (black). At this stage, many of the nephrons cannot be recovered anymore. Currently, biopsy is performed to estimate the underlying CKD etiology and to guide the therapy. Nevertheless, the information about the ideal (personalized) treatment method can generally not be extracted from the pathological examination (question mark). This may result in the application of drugs that do not or only moderately interfere with CKD progression. The application of non-invasive biomarkers holds the promise to allow for early diagnosis based on molecular changes, before irreversible damage has occurred. Moreover, these biomarkers should distinguish different CKD etiologies, and, even more important for the patients, should enable prediction of treatment response, thereby supporting the definition of the best suited, personalized treatment.

**Figure 2 ijms-25-03519-f002:**
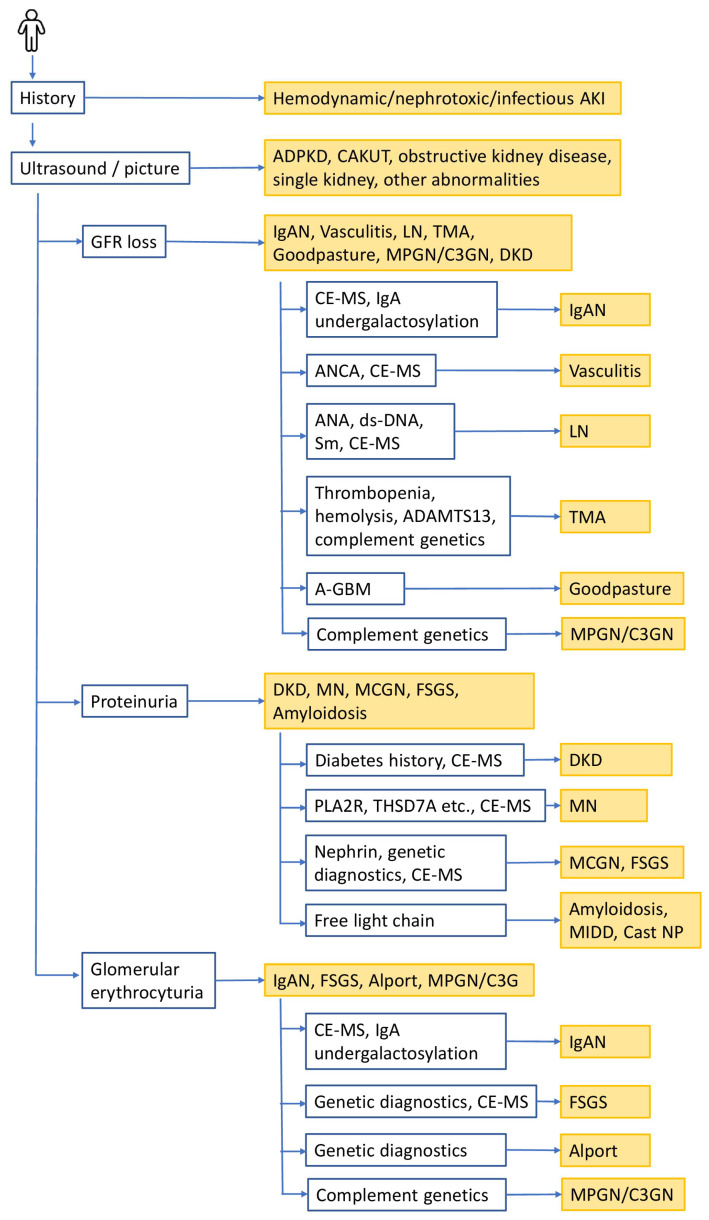
Proposed decision tree for a non-invasive biomarker-guided diagnostic path in chronic kidney disease (CKD) in cases where conventional kidney biopsy is not feasible. Based on the literature, current available prognostic and predictive biomarkers for the management of CKD were combined to provide guidance on the application of these biomarkers. Given are the available biomarkers with the application for the specific disease. If diagnosis cannot be established with confidence based on the presented decision tree, additional biomarkers, especially in the context of rare disease may be considered, depending on the clinical presentation.

**Table 1 ijms-25-03519-t001:** Overview of biomarkers for glomerular diseases.

Disease	Biomarker	Sample	Application	Routine Clinical Use
IgAN	Gd-IgA1	Serum	Diagnosis; prognosis; disease monitoring	No
IgAN	DKK3	Urine	Prognosis	No
IgAN	IgAN237	Urine	Prognosis	Yes
MCGN	Anti-nephrin AB	Serum	Diagnosis; treatment response	No
MCGN	CD80	Serum/Urine	Diagnosis; relapse detection	No
FSGS	suPAR	Serum	Diagnosis; disease monitoring	No
FSGS	ApoA-Ib	Urine	Relapse detection	No
MN	PLA2R1	Serum	Diagnosis	Yes
MN	THSD7A	Serum	Diagnosis	Yes
MN	SEMA3B	Serum	Diagnosis; malignancy	No
MN	Nell-1	Serum	Diagnosis	Yes
MPGN/C3GP	Complement factors	Serum	Diagnosis; prognosis; disease monitoring	Yes
FGN	DNAJB9	Serum	Diagnosis	No
AAV	ANCA	Serum	Diagnosis	Yes
AAV	Calprotectin	Serum	Relapse risk	No
AAV	MCP-1	Urine	Disease monitoring	No
AAV	sCD163	Urine	Diagnosis; relapse detection; disease monitoring	No
LN	Anti-C1q	Serum	Diagnosis; disease monitoring	No
LN	MCP-1	Urine	Diagnosis; prognosis; disease monitoring	No
LN	EGF	Urine	Prognosis	No
LN	KIM-1	Urine	Disease monitoring; prognosis	No
LN	NGAL	Urine	Disease monitoring; prognosis	No
DKD	TNFR 1 and 2	Serum	Prognosis	No
DKD	CRP	Plasma	Prognosis	No
DKD	KIM-1	Plasma/Urine	Prognosis	No
DKD	NGAL	Urine	Diagnosis	No
DKD	CKD273	Urine	Prognosis	Yes

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
