# Peer review of "Non-Invasive Biomarkers for Diagnosis, Risk Prediction, and Therapy Guidance of Glomerular Kidney Diseases: A Comprehensive Review"

_ijms, 2024, doi:10.3390/ijms25063519_

Round 1

Reviewer 1 Report

Comments and Suggestions for Authors

This manuscript represents an attempt to undertake a review on what is currently known about non invasive biomarkers in the setting of glomerular renal disease.

There are some problems with the manuscript in its current form. They are as follows-

1) There are a number of sweeping statements in the abstract. The first sentence possibly overstates the importance of glomerular kidney diseases as a significant health problem internationally. If a statement such as this is to be made, then it must be supported by data. The second is the statement that for the large majority of patients with glomerular kidney diseases valid biomarkers are available. This may be true to some extent but is not completely true around the world (where in many places there is limited access to medical and/or nephrology services-which is worse in some regions right now due to the various conflicts). The authors are best to focus on what the results of the review have actually revealed.

2) Towards the end of the Introduction section mention is made of the focus being on urine peptidomics, however this is not the case throughout Section 2, where both blood and urine biomarkers are covered. Plus, the word peptidomics may not be the correct term to be using for this particular manuscript.

3) Are there any published guidelines on the use of urinary biomarkers in any of these subgroups of patient? Plus, apart from the urinary biomarkers having been studied in research settings, what is known about their uptake in clinical practice? Have the regulatory authorities in either North America or Europe approved any commercial assays and if so which ones? Have these particular assays all been adequately prospectively validated, or is there ongoing work in that space? Are any of these urinary biomarkers available as a panel and if so in what settings are the panels being used? Is cost an issue when it comes to more widespread adoption of these assays?

4) The reason all of the questions in point 3 were raised is because the manuscript contains a proposed decision tree for a non-invasive biomarker guided diagnostic path in chronic kidney disease. The problem with this proposed decision tree is that it is not clear if there is enough evidence to support it being widely adopted in practice. Plus there is the issue of all of the relevant tests being readily and widely accessible as well as the cost. Hence, the authors need to provide more in the way of information as to how it could be adopted, precisely when it would actually be used and in what possible clinical settings. Is it not the case that more ongoing research is actually required?

Author Response

Response to Reviewer 1 Comments

General comment:

This manuscript represents an attempt to undertake a review on what is currently known about non invasive biomarkers in the setting of glomerular renal disease.

There are some problems with the manuscript in its current form. They are as follows-

Comment 1: There are a number of sweeping statements in the abstract. The first sentence possibly overstates the importance of glomerular kidney diseases as a significant health problem internationally. If a statement such as this is to be made, then it must be supported by data. The second is the statement that for the large majority of patients with glomerular kidney diseases valid biomarkers are available. This may be true to some extent but is not completely true around the world (where in many places there is limited access to medical and/or nephrology services-which is worse in some regions right now due to the various conflicts). The authors are best to focus on what the results of the review have actually revealed.

Response: We thank the reviewer for the valuable comment and revised the abstract section as followed: “Effective management of glomerular kidney disease, one of the main categories of chronic kidney disease (CKD), requires accurate diagnosis, prognosis of progression, assessment of therapeutic efficacy, and, ideally, prediction of drug response. Multiple biomarkers and algorithms for the assessment of specific aspects of glomerular diseases have been reported in the literature. Though, the vast majority of these have not been implemented in clinical practice or are not available on a global scale due to limited access, missing medical infrastructure or economical as well as political reasons.“

Comment 2: Towards the end of the Introduction section mention is made of the focus being on urine peptidomics, however this is not the case throughout Section 2, where both blood and urine biomarkers are covered. Plus, the word peptidomics may not be the correct term to be using for this particular manuscript.

Response: We agree with the review and revised the section of the introduction in order not to give misleading information:

“Here we review current data on non-invasive biomarkers to diagnose specific glomerular diseases and predict progression. We discuss data on non-invasive biomarkers in the context of specific glomerular diseases, focusing on urine proteomic and peptidomic biomarkers, as this approach offers the opportunity to develop a non-invasive and unbiased diagnostic tool without a priori assumptions as to the pathogenesis of a disease. Urine has several advantages compared to plasma or serum, among others reduced complexity and increased stability. We aim to give a complete collection of blood and urine biomarkers of pathophysiological aspects of CKD and nine different categories of glomerular diseases. We refer to the most widely accepted, validated and available markers in this review. A comprehensive over-view of these markers is given in Supplementary Table 1.”

Comment 3: Are there any published guidelines on the use of urinary biomarkers in any of these subgroups of patient? Plus, apart from the urinary biomarkers having been studied in research settings, what is known about their uptake in clinical practice? Have the regulatory authorities in either North America or Europe approved any commercial assays and if so which ones? Have these particular assays all been adequately prospectively validated, or is there ongoing work in that space? Are any of these urinary biomarkers available as a panel and if so in what settings are the panels being used? Is cost an issue when it comes to more widespread adoption of these assays?

Comment 4: The reason all of the questions in point 3 were raised is because the manuscript contains a proposed decision tree for a non-invasive biomarker guided diagnostic path in chronic kidney disease. The problem with this proposed decision tree is that it is not clear if there is enough evidence to support it being widely adopted in practice. Plus there is the issue of all of the relevant tests being readily and widely accessible as well as the cost. Hence, the authors need to provide more in the way of information as to how it could be adopted, precisely when it would actually be used and in what possible clinical settings. Is it not the case that more ongoing research is actually required?

Response to comment 3 and 4: The questions raised in the comments 3 and 4 are very important. To our knowledge only a minority of biomarkers are incorporated by society guidelines and the current state of biomarker research for the majority of glomerular diseases is still in an early phase and promising candidates which we propose in this manuscript need further investigation and ultimately validation in larger, independent and specific cohorts which need to be defined. We added a paragraph dedicated to this issue including biomarkers that already have enough evidence to support a decision to not perform a kidney biopsy, for example AAV and MN. To support this claim, we also added the corresponding KDIGO guideline. As for the rest of the third comment, the majority of the biomarkers has not been prospectively validated. Cost is always an issue when it comes to clinical implementation. Ultimately, we believe that if a biomarker can replace a kidney biopsy or add prognostic value or information on treatment response which the kidney biopsy cannot in certain cases, cost efficiency will be much higher with the specific biomarker. Even more so if methods like LC/MS become more widely available and cost efficiency is constantly improved.

“Current standards in diagnosis and management of glomerular diseases rely heavily on invasive kidney biopsy and KDIGO guidelines for glomerular diseases (https://kdigo.org/wp-content/uploads/2017/02/KDIGO-Glomerular-Diseases-Guideline-2021-English.pdf) revolve mainly around histological information. Regarding the current magnitude and speed of non-invasive biomarker research and considering some examples of glomerular diseases (MN, AAV) where non invasive biomarkers have in part replaced kidney biopsy, we believe that the role of invasive diagnostic tools like kidney biopsy could become smaller due to risks and costs of the procedure in case that robust and reliable tools based on non-invasive diagnostic become widely available. We therefore want to provide a theoretical decision-tree including current research findings for scenarios where kidney biopsy might not be possible or feasible.”

Reviewer 2 Report

Comments and Suggestions for Authors

This review article describes  non-invasive biomarkers for diagnosis, risk prediction and  guiding therapy of glomerular kidney diseases.

The article is interesting and well-written.

It would have been interesting if the authors had discussed in conclusion the directions of research on the search for new diagnostic markers in glomerular kidney diseases, especially those that could find application in daily clinical practice.

A table summarizing markers in blood and urine in various glomerular kidney diseases would also be helpful.

Comments on the Quality of English Language

Minor editing of English language required

Author Response

Response to Reviewer 2 Comments

General comment:

This review article describes non-invasive biomarkers for diagnosis, risk prediction and guiding therapy of glomerular kidney diseases.

The article is interesting and well-written.

Response: We thank the reviewer for this comment.

Comment 1: It would have been interesting if the authors had discussed in conclusion the directions of research on the search for new diagnostic markers in glomerular kidney diseases, especially those that could find application in daily clinical practice.

Response: We thank the reviewer for this remark. We added a conclusion paragraph mentioning future perspectives for biomarker research in glomerular diseases:

“4. Conclusion

In this review we give an overview of biomarkers of glomerular kidney diseases which have been implemented in clinical practice or are promising candidates for future clinical implementation. We provide a clinical decision-tree for non-invasive diagnosis and management of glomerular diseases in cases of kidney biopsy not being available or feasible. Most of the biomarkers are not yet incorporated in KDIGO guidelines on management of glomerular diseases, in some cases independent validation in larger cohorts is missing. Due to the downsides of invasive kidney biopsy the need for biomarkers in glomerular diseases is high and encouraged by KDIGO. Future biomarker research using large-sclae approaches like proteomics, transcriptomics and metabolomics seem to be most promising. Furthermore, untargeted approaches and unsupervised machine-learning using AI might offer new perspectives to biomarker panels or classifications of specific glomerular diseases like MCD or FSGS.”

Comment 2: A table summarizing markers in blood and urine in various glomerular kidney diseases would also be helpful.

Response: We thank the reviewer for this remark. Supplementary table 1 shows all included glomerular biomarkers but was too long to include in the manuscript. We added a table with a reduced number of biomarkers and essential information to the manuscript.

Table 1: Overview of biomarkers for glomerular diseases

Disease

Biomarker

Sample

Application

Routine clinical use

IgANP

Gd-IgA1

Serum

Diagnosis; Prognosis; Disease Monitoring

No

IgANP

DKK3

Urine

Prognosis

No

IgANP

IgAN237

Urine

Prognosis

No

MCD

Anti-nephrin AB

Serum

Diagnosis; Treatment response

No

MCD

CD80

Serum/Urine

Diagnosis; Relapse detection

No

FSGS

suPAR

Serum

Diagnosis; Disease monitoring

No

FSGS

ApoA-Ib

Urine

Relapse detection

No

MN

PLA2R1

Serum

Diagnosis

Yes

MN

THSD7A

Serum

Diagnosis

Yes

MN

SEMA3B

Serum

Diagnosis; Malignancy

No

MN

Nell-1

Serum

Diagnosis

Yes

MPGN/C3GP

Complement factord

Serum

Diagnosis; Prognosis; Disease Monitoring

Yes

FGN

DNAJB9

Serum

Diagnosis

No

AAV

ANCA

Serum

Diagnosis

Yes

AAV

Calprotectin

Serum

Relapse risk

No

AAV

MCP-1

Urine

Disease monitoring

No

AAV

sCD163

Urine

Diagnosis; Relapse detection; Disease monitoring

No

LN

Anti-C1q

Serum

Diagnosis; Disease monitoring

No

LN

MCP-1

Urine

Diagnosis; Prognosis; Disease Monitoring

No

LN

EGF

Urine

Prognosis

No

LN

KIM-1

Urine

Disease monitoring; Prognosis

No

LN

NGAL

Urine

Disease monitoring; Prognosis

No

DKD

TNFR 1 and 2

Serum

Prognosis

No

DKD

CRP

Plasma

Prognosis

No

DKD

KIM-1

Plasma/Urine

Prognosis

No

DKD

NGAL

Urine

Diagnosis

No

DKD

CKD273

Urine

Prognosis

No

Reviewer 3 Report

Comments and Suggestions for Authors

The present review reports a current guide of diagnostic, prognostic and predictive biomarkers available for the management of glomerular diseases and algorithms for the assessment of specific aspects of these diseases.

The language seems correct throughout the text and the references are appropriate, adequate and up-to-date.

However, several tubular injury biomarkers, inflammatory biomarkers, biomarkers of fibrosis, and biomarkers of oxidative stress could be included. TGF-β through the signalling pathway of intracellular proteins that transduce extracellular signals namelly small mother against decapentaplegic (SMADs) regulates VEGF. VEGF is a key angiogenic factor, influencing the proliferation of endothelial cells and plays a pivotal role in vascular integrity, pathological angiogenesis and has been implicated in diabetic nephropathy and chronic glomerulonephritis. Biomarkers of glomerular injury as immunoglobulin G (IgG4 & 2 isoforms), ceruloplasmin, collagen type IV, laminin, fibronectin, podocytes-podocalyxin, glycosaminoglycans (GAGs), licalin-type prostaglandin D synthase (L-PGDS), and micro-RNAs could also be mentioned. All the prevous approaches and prompts are aimed at the fuller approximation of this important clinical model.

1. The main question addressed by the research is to provide a complete and well structured guide of glomerular diseases biomarkers.

2. The structure of the guide must be more extensive and complete including growth factors of chronic kidney disease (TGF-β1, FGF-23, VEGF-A...) and col-IV molecules and basement membane degradation, important in glomerular disease process initiation and evolution.

3. The present review should aspire to become a point of reference of the bibliography, overshadowing other in the literature with the same object.

4. An examplary model described to the initial comments could include biomarkers of tubula injury, biomarkers of inflammation and oxidative stress  (as they represent the main pathways of endothelial dysfunction).

5. Conclusions are accurate and comprehensive, consistent with the evidence and arguments presented. In an overall effort to improve we must know that the enemy of the good is thw best.

6. References are appropriate, adequate and up-to-date.

7. Biomarkers and algorithms for the assessment of specific aspects of glomerular diseases must be more extensive presented and better organized and structured.

Author Response

Response to Reviewer 3 Comments

General comment:

The present review reports a current guide of diagnostic, prognostic and predictive biomarkers available for the management of glomerular diseases and algorithms for the assessment of specific aspects of these diseases.

The language seems correct throughout the text and the references are appropriate, adequate and up-to-date.

Response: We are pleased to hear that the reviewer is satisfied with these aspects of the manuscript.

However, several tubular injury biomarkers, inflammatory biomarkers, biomarkers of fibrosis, and biomarkers of oxidative stress could be included. TGF-β through the signalling pathway of intracellular proteins that transduce extracellular signals namelly small mother against decapentaplegic (SMADs) regulates VEGF. VEGF is a key angiogenic factor, influencing the proliferation of endothelial cells and plays a pivotal role in vascular integrity, pathological angiogenesis and has been implicated in diabetic nephropathy and chronic glomerulonephritis. Biomarkers of glomerular injury as immunoglobulin G (IgG4 & 2 isoforms), ceruloplasmin, collagen type IV, laminin, fibronectin, podocytes-podocalyxin, glycosaminoglycans (GAGs), licalin-type prostaglandin D synthase (L-PGDS), and micro-RNAs could also be mentioned. All the prevous approaches and prompts are aimed at the fuller approximation of this important clinical model.

Comment 1: The main question addressed by the research is to provide a complete and well structured guide of glomerular diseases biomarkers.

Comment 2: The structure of the guide must be more extensive and complete including growth factors of chronic kidney disease (TGF-β1, FGF-23, VEGF-A...) and col-IV molecules and basement membane degradation, important in glomerular disease process initiation and evolution.

Comment 3: The present review should aspire to become a point of reference of the bibliography, overshadowing other in the literature with the same object.

Comment 4: An examplary model described to the initial comments could include biomarkers of tubula injury, biomarkers of inflammation and oxidative stress (as they represent the main pathways of endothelial dysfunction).

Response to comments 1 to 4: We thank the reviewer for this extensive comment which we believe is very important. In order to limit the word count of this manuscript we already had to limit the number of biomarkers to what we believe are the most promising ones for clinical implementation regarding glomerular diseases. Certainly, the above-mentioned biomarkers are highly relevant for the understanding of the mechanisms revolving around the pathophysiological aspects of kidney injury, inflammation and scarring namely fibrosis which also play a key role in all glomerular diseases. The above-mentioned biomarkers and processes are included in our accompanying manuscript entitled “Assessment and Risk Prediction of Chronic Kidney Diseases and Kidney Fibrosis Using Non-Invasive Biomarkers” (ijms-2887341) as they are relevant in damage mechanisms evolving around CKD in general and fibrosis. In this manuscript we aimed to include only disease specific biomarkers of glomerular diseases that might have implications for differential diagnosis, estimating prognosis or guiding therapeutic strategies. We hope the reviewer can agree with our assessment while we value his comment greatly.

Comment 5: Conclusions are accurate and comprehensive, consistent with the evidence and arguments presented. In an overall effort to improve we must know that the enemy of the good is the best.

Comment 6: References are appropriate, adequate and up-to-date.

Response to comment 5 and 6: We thank the reviewer for these comments.

Comment 7: Biomarkers and algorithms for the assessment of specific aspects of glomerular diseases must be more extensive presented and better organized and structured.

Response: We thank the reviewer for this remark. We made several changes to the manuscript including a table overviewing the most important biomarkers and a conclusion in order to enhance organization and structure.

Round 2

Reviewer 1 Report

Comments and Suggestions for Authors

The authors have adequately responded to the reviewers comments

Reviewer 3 Report

Comments and Suggestions for Authors

Published in its current form.